# The cytokine GDF15 signals through a population of brainstem cholecystokinin neurons to mediate anorectic signalling

Amy A Worth[1], Rosemary Shoop[1], Katie Tye[1], Claire H Feetham[1], Giuseppe D'Agostino[1,2], Garron T Dodd[3], Frank Reimann[4], Fiona M Gribble[4], Emily C Beebe[5], James D Dunbar[5], Jesline T Alexander-Chacko[5], Dana K Sindelar[5], Tamer Coskun[5], Paul J Emmerson[5], Simon M Luckman[1]*

[1]Faculty of Biology, Medicine and Health, The University of Manchester, Manchester, United Kingdom; [2]Rowett Institute, University of Aberdeen, Aberdeen, United Kingdom; [3]School of Biomedical Sciences, The University of Melbourne, Victoria, Australia; [4]Institute of Metabolic Science, Addenbrooke's Hospital, Cambridge, United Kingdom; [5]Lilly Research Laboratories, Eli Lilly & Company, Indianapolis, United States

**Abstract** The cytokine, GDF15, is produced in pathological states which cause cellular stress, including cancer. When over expressed, it causes dramatic weight reduction, suggesting a role in disease-related anorexia. Here, we demonstrate that the GDF15 receptor, GFRAL, is located in a subset of cholecystokinin neurons which span the area postrema and the nucleus of the tractus solitarius of the mouse. GDF15 activates GFRAL[AP/NTS] neurons and supports conditioned taste and place aversions, while the anorexia it causes can be blocked by a monoclonal antibody directed at GFRAL or by disrupting CCK neuronal signalling. The cancer-therapeutic drug, cisplatin, induces the release of GDF15 and activates GFRAL[AP/NTS] neurons, as well as causing significant reductions in food intake and body weight in mice. These metabolic effects of cisplatin are abolished by pre-treatment with the GFRAL monoclonal antibody. Our results suggest that GFRAL neutralising antibodies or antagonists may provide a co-treatment opportunity for patients undergoing chemotherapy.

*For correspondence:
simon.luckman@manchester.ac.uk

## Introduction

The cytokine, GDF15 (a member of the TGF-β cytokine family, also known as MIC-1 and NAG-1), is expressed in several tissues throughout the body and circulates in the bloodstream of healthy humans (*Bootcov et al., 1997*; *Tsai et al., 2018*; *Patel et al., 2019*). Plasma levels increase dramatically in a number of pathological states associated with cellular stress, including cancers, cardiac failure, chronic kidney disease, infection and obesity (*Patel et al., 2019*; *Welsh et al., 2003*; *Kempf et al., 2007*; *Ho et al., 2013*; *Bauskin et al., 2006*; *Luan et al., 2019*). Furthermore, over expression of GDF15 causes a dramatic reduction in food intake and weight loss (*Tsai et al., 2018*; *Johnen et al., 2007*; *Macia et al., 2012*; *Chrysovergis et al., 2014*; *Xiong et al., 2017*). Together, this has led to the supposition that GDF15 does not have a normal, physiological role but is, instead, secreted as an adaptive response to disease (*Patel et al., 2019*; *Luan et al., 2019*). This said, the transgenic knock out of GDF15 from the germline results in obesity, which could be interpreted that the cytokine has an alternative physiological function to regulate body weight (*Tsai et al., 2013*; *Low et al., 2017*; *Tran et al., 2018*).

The GDF15 receptor, GFRAL (GDNF-family receptor α-like) is located exclusively in a small population of cells in the in the area postrema (AP) and nucleus of the tractus solitarius (NTS) of the

mouse dorsomedial medulla oblongata, (*Mullican et al., 2017*; *Yang et al., 2017*; *Emmerson et al., 2017*; *Hsu et al., 2017*) a brainstem region containing a number of characterised neurons that previously have been linked with appetite regulation (*Luckman, 1992*; *Rinaman et al., 1993*; *Larsen et al., 1997*; *Lawrence et al., 2000*; *Luckman and Lawrence, 2003*; *Ellacott et al., 2006*; *D'Agostino et al., 2016*; *Roman et al., 2016*; *Frikke-Schmidt et al., 2019*). Administration of recombinant GDF15 reduces food intake, but not in GFRAL knock-out mice (*Mullican et al., 2017*; *Yang et al., 2017*; *Emmerson et al., 2017*; *Hsu et al., 2017*). It also induces the cellular activation marker, Fos protein, in GFRAL-positive cells in the AP/NTS, and in putative downstream targets in the pons and amygdala, (*Xiong et al., 2017*; *Hsu et al., 2017*; *Frikke-Schmidt et al., 2019*), whereas selective surgical lesioning of the AP/NTS blocks the anorexic effects of the peptide (*Tsai et al., 2014*). The absolute identity of the primary responsive neurons has not been determined. Although a small number of GFRAL-positive neurons contain immunoreactivity for the catecholaminergic marker, tyrosine hydroxylase (TH), (*Yang et al., 2017*) relatively few TH-positive neurons are activated by exogenous GDF15 (*Tsai et al., 2014*). We have extended these investigations and found that the highest proportion of GFRAL-positive neurons contain the neuropeptide transmitter, cholecystokinin (CCK). We demonstrate that GFRAL cells are a sub-population of CCK neurons, which respond to administration of GDF15 or the cancer therapeutic drug, cisplatin, but not to other anorectic signals. Additionally, the effect of GDF15 to inhibit food intake is abrogated by the targeted deletion of CCK-containing neurons in the AP/NTS or by pre-administration of a CCK receptor antagonist. A single injection of GDF15 causes marked conditioned taste and place aversions, suggesting a strong negative affect, as well as activating downstream pathways previously described as mediating anorexia. Lastly, since we can block GDF15- or cisplatin-induced anorexia by neutralising the GFRAL receptor with a selective monoclonal antibody, we suggest that the anorectic responses to disease and, potentially, their therapeutic treatment may be mediated by this distinct signalling pathway.

## Results

### GFRAL is localised to CCK-positive neurons in the AP/NTS

Mice expressing Cre recombinase under the control of neuropeptide genes, were crossed with a reporter mouse expressing enhanced Yellow Fluorescent Protein in a Cre-dependent fashion (Rosa26-eYFP), so that the double mutants expressed eYFP in discrete populations of CCK (*Taniguchi et al., 2011*), glutamate (VGlut2; gene *Slc17a6*), preproglucagon (PPG; gene *Gcg*) (*Parker et al., 2012*) or prolactin-releasing peptide (PrRP; gene *Prlh*) (*Dodd et al., 2014*) neurons in the AP/NTS. Using antibodies against GFRAL, eYFP and TH, we were able to characterise putative, GDF15-sensitive cells. In the mouse, GFRAL was expressed in a continuous grouping of cells in the AP, extending into the medial region of the NTS (*Figure 1A*), as noted previously by others (*Mullican et al., 2017*; *Yang et al., 2017*; *Emmerson et al., 2017*; *Hsu et al., 2017*). Although dense within the AP and more sparsely distributed in the NTS, there were similar numbers of GFRAL neurons in the two structures (20 ± 1 per section in both the AP and the NTS). 60% of GFRAL-immunoreactive cells in the AP co-localised with $Cck^{Cre}$::eYFP, though this proportion was 31% in the NTS (*Figure 1B* and summary in *Supplementary file 1*). We confirmed this pattern using RNAScope in situ hybridisation histology, which provided slightly higher values of 69% and 35% overlap in the AP and NTS, respectively (*Figure 1—figure supplement 1A* and *Supplementary file 1*). Thus, overall, the majority of GFRAL neurons are CCKergic. By comparison, dual immunohistochemistry revealed that 27% of GFRAL neurons in the AP and 45% in the NTS contained TH (*Figure 1B*). Overall, approximately, 15% of GFRAL neurons contained both CCK and TH (*Figure 1—figure supplement 1B*). At least 54% GFRAL neurons in the AP co-localised with $Slc17a6^{Cre}$::eYFP, which fits with the consensus that CCK neurons in this brain region are glutamatergic (*Figure 1—figure supplement 1C*). Finally, GFRAL cells were distinct from other NTS populations which contain either PrRP or PPG (*Figure 1B*), which themselves either form a separate sub-population of TH neurons (*Dodd and Luckman, 2013*) or overlap with other CCK cells (*Garfield et al., 2012*; *Figure 1—figure supplement 1D*), respectively. In conclusion, GFRAL cells appear to form a distinct subset of CCK neurons, a proportion of which also contain TH.

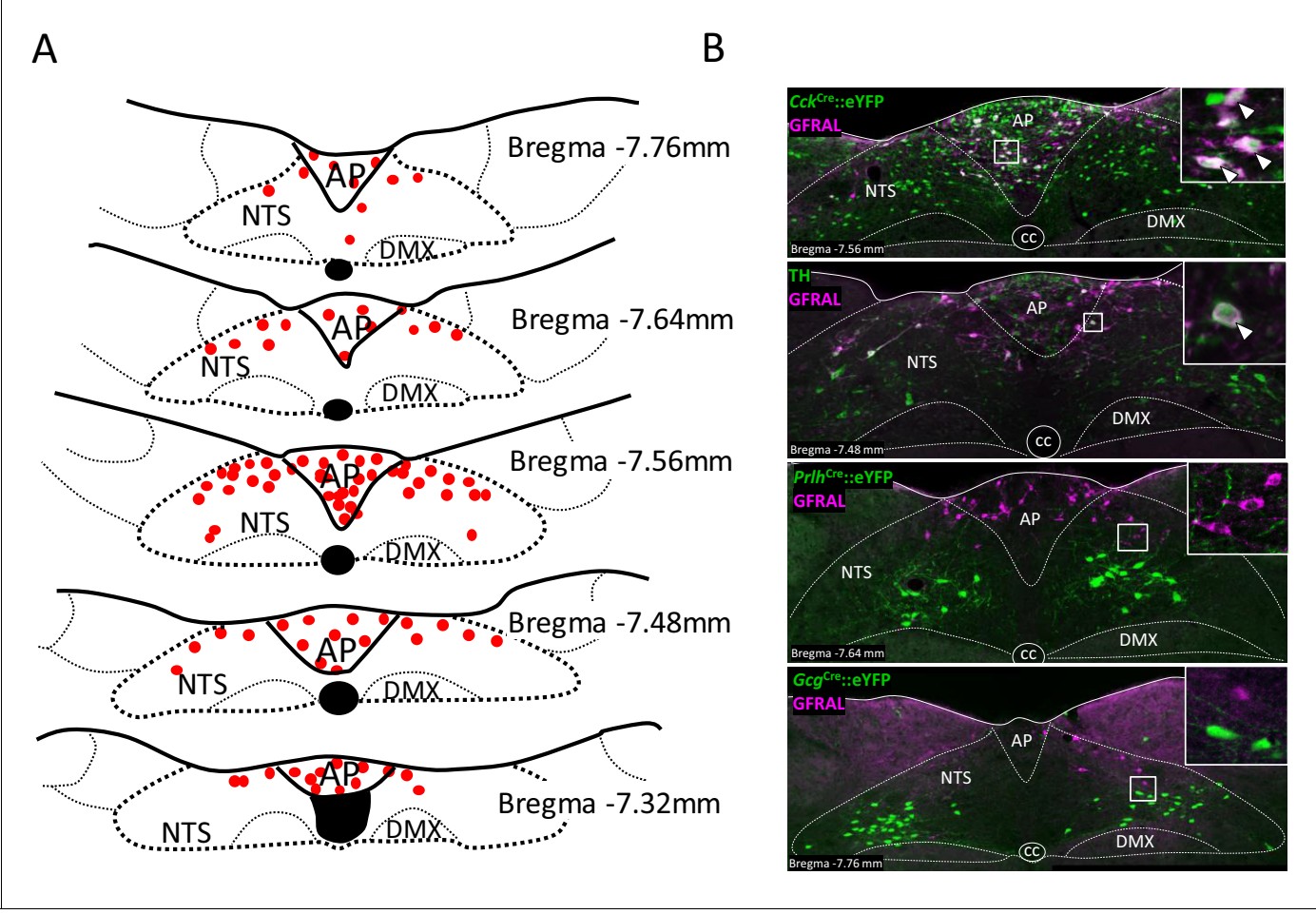

**Figure 1.** GFRAL-positive neurons in the AP and NTS co-localise with CCK. (**A**) Schematic describing the distribution of GFRAL-immunoreactive cell bodies in the AP and dorsal NTS at different rostrocaudal levels relative to *bregma*. (**B**) Dual-fluorescence labelling for GFRAL (magenta) with TH or eYFP (staining using antibody raised against green fluorescent protein) in three reporter mice, *Cck*^Cre::eYFP, *Prlh*^Cre::eYFP or *Gcg*^Cre::eYFP. GFRAL co-localised with CCK and TH, but not PrRP or PPG (the latter being located more caudal to the majority of GFRAL neurons). White arrows in higher magnification inset indicate co-labelled cells. AP (area postrema), cc (central canal), DMX (dorsal motor nucleus of the tenth cranial nerve, vagus), NTS (nucleus of the tractus solitarius).

The online version of this article includes the following figure supplement(s) for figure 1:

**Figure supplement 1.** Further histological analysis of GFRAL/CCK neurons.

## GDF15 produces anorexia and a negative affective valence

A single injection of GDF15 (2–8 nmol/kg, subcutaneous; s.c.) at lights-out produced a significant and dose-dependent decrease in normal, night-time feeding within 2 hr (hr) of administration (*Figure 2A*). Cumulative food intake had mostly recovered by 24 hr. This is within the range used by others who report dose-dependent effects of single, systemic injections of GDF15 in mice (*Patel et al., 2019*; *Mullican et al., 2017*). Doses of 4–8 nmol/kg were required to see a significant effect on fast-induced, day-time feeding (*Figure 2B*), and were the doses used in later experiments. Administration of GDF15 was associated with a negative affective valence, since a single injection supported a strong conditioned taste aversion (CTA) when paired with sucrose (see also *Patel et al., 2019*), and a conditioned place aversion (CPA) in mice (*Figure 2C and D*). In addition, pica behaviour, in which kaolin clay is consumed to remedy gastric malaise, was used as a measure of sickness given that rodents cannot vomit. Administration of GDF15 to rats on 3 consecutive days induced pica behaviour – a proxy for sickness behaviour – to an extent comparable to that seen following injection of the nausea-inducing toxin, lithium chloride (LiCl; *Figure 2—figure supplement 1*).

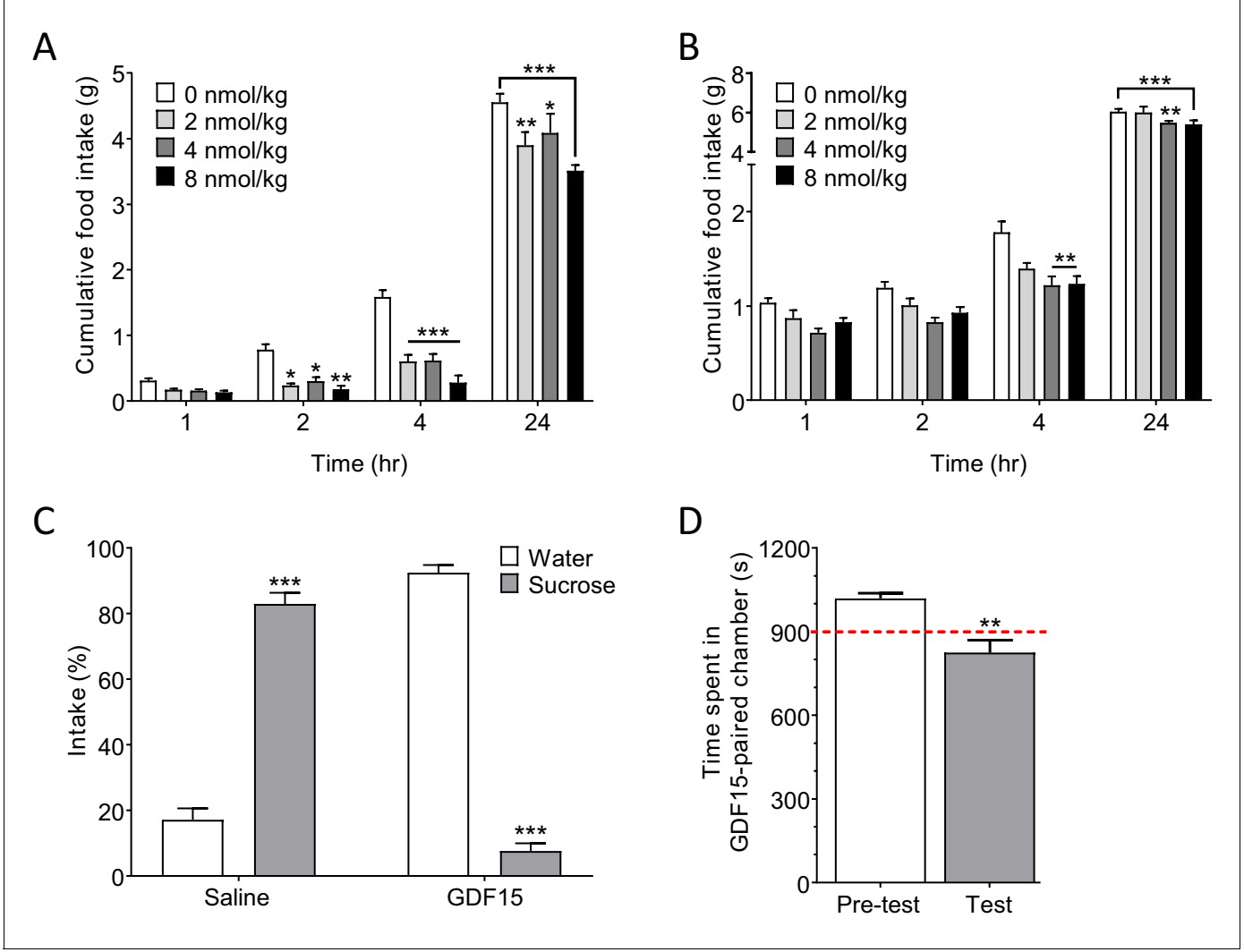

**Figure 2.** GDF15 produces anorexia and a negative affective valence. (**A**) Subcutaneous administration of GDF15, just before 'lights out,' decreased normal, night-time feeding (n = 6 per group; *p<0.05, **p<0.01, ***p<0.001, compared with 0 nmol/kg group; two-way ANOVA followed by Tukey's multiple comparison test). (**B**) GDF15 also decreased fast-induced, day-time re-feeding (n = 5–6 per group; **p<0.01, ***p<0.001, compared with 0 nmol/kg group). (**C**) GDF15 supported a conditioned taste aversion in mice when paired with sucrose as the conditioned stimulus. Data show two-bottle fluid intake 24 hr following a single conditioning to GDF15 (n = 6 per group; ***p<0.001, water versus sucrose intake for saline- and GDF15-treated groups; two-way ANOVA followed by Sidak's multiple comparison test). (**D**) GDF15 supported a conditioned place aversion in mice. Mice showed a preference for one side of the arena measured as time spent (seconds) in preferred side. During conditioning, mice received an injection of GDF15 on their preferred (dark) side and saline on their non-preferred side. On the test day, the mice displayed a decreased preference for the side on which they received GDF15 (n = 12; **p<0.01, time spent in preferred side; paired t-test).

The online version of this article includes the following figure supplement(s) for figure 2:

**Figure supplement 1.** GDF15 supports sickness behaviour in rats.

Together, these and recently published data (*Borner et al., 2020a*; *Borner et al., 2020b*) suggest that GDF15 is probably not a natural satiety factor, but exerts a pathophysiological action to cause anorexia. This conclusion is supported by the findings that circulating GDF15 levels do not correlate with meal times in humans, (*Patel et al., 2019*; *Tsai et al., 2015*) and that GDF15 knock out in mice does not result in significant changes in normal chow intake (*Tsai et al., 2013*; *Tran et al., 2018*).

## GDF15 activates CCK[AP/NTS] neurons

Next, we determined the identity of GDF15-activated neurons by carrying out Fos-activity mapping in *Cck*[Cre]::eYFP, *Prlh*[Cre]::eYFP, *Gcg*[Cre]::eYFP or *Pomc*[eGFP] mice. A single, low-anorectic dose of GDF15 activated GFRAL+ve/CCK and TH neurons in the AP/NTS, as well as GFRAL-ve neurons at the same level of both the AP and NTS (*Figure 3A*; *Figure 3—figure supplement 1A and B*). Neither PrRP, PPG nor POMC neurons are activated significantly by GDF15 (*Figure 3—figure supplement 1B*). Thus, a very obvious group of GFRAL-ve cells in the medial NTS, that are activated by GDF15, remain unidentified (*Figure 3A*, arrow head). GFRAL cells are not activated by natural satiety signals acting after meal intake (*Figure 3—figure supplement 1C*); that is, following fast-induced re-feeding, the infusion of lipid directly into the stomach (300 µl Intralipid by gavage) or a low-dose of satiety-inducing, systemic CCK (6 µg/kg body weight, intraperitoneal; i.p.). Perhaps surprisingly, only a small number are activated by LiCl (128 mg/kg body weight, i.p.). This contrasts with the general activation profile of CCK neurons in the AP/NTS, which respond to different anorectic stimuli and which underlines that they are probably a mixed population (*D'Agostino et al., 2016*; *Roman et al., 2017*).

In terms of potential downstream mediators of the GDF15 signal, in addition to the non-GFRAL cells in the NTS and AP, significant increases in Fos staining were recorded in the lateral parabrachial nucleus of the pons (PBN), the paraventricular nucleus of the hypothalamus (PVH), the oval sub-nucleus of the bed nucleus of the stria terminalis (ovBNST) and in the central nucleus of the amygdala (CeA; *Figure 3—figure supplement 1D*). Using *Calca*[Cre]::eYFP or *Crh*[Cre]::eYFP mice, we show that some neurons in the PBN activated by GDF15 express calcitonin gene-related peptide (CGRP), while many in the PVH, but not the CeA and ovBNST, express corticotrophin-releasing hormone (CRH; *Figure 3B* and *Figure 3—figure supplement 1E*). Instead, within the ovBNST and CeA, a large proportion of Fos-positive cells contained PKC-δ immunoreactivity (*Figure 3B*). CCK[NTS] neurons project directly to the PBN, including to cells containing CGRP (*Roman et al., 2016*; *Roman et al., 2017*); and, CCK$_1$ receptors in the PBN have been proposed to modulate information flow from gut to brain (*Mercer and Beart, 2004*; *Saleh et al., 1997*). CGRP[PBN] neurons respond to a number of anorectic signals and may act as a point of convergence for different signalling pathways, themselves projecting forward to the CeA and elsewhere (*Wu et al., 2012*; *Carter et al., 2013*). Therefore, we used dual-fluorescence RNAScope to demonstrate the expression of *Cckr1* in the lateral PBN, but found the receptor mRNA in relatively few CGRP (*Calca* mRNA-expressing) cells (*Figure 3—figure supplement 1F*). Thus, CGRP neurons are unlikely to be the only target in the PBN for GFRAL neurons.

By repeating our Fos experiment but in mice previously injected with the retrograde tracer, Fluoro-Gold, into the lateral PBN, we demonstrate that both GFRAL and CCK neurons activated by GDF15 project directly to the PBN (*Figure 3C*). Likewise, CCK[NTS] neurons also send direct projections to the PVH (*D'Agostino et al., 2016*). We confirmed this projection pattern using retrograde tracing, however, we showed that almost no GDF15-activated, GFRAL+ve or GFRAL-ve AP/NTS cells project directly to the PVH or to the ovBNST (*Figure 3—figure supplement 1G and H*). The most parsimonious conclusion is that GFRAL cells activated by GDF15 project directly to the PBN, which then activates downstream targets in the CeA, ovBNST and PVH. GFRAL cells may also synapse locally to activate other neuronal populations, including cells in the medial NTS. A very small number of these synaptically activated cells contain CCK or TH, and because almost none contain either PrRP, PPG or POMC, they may represent another distinct NTS phenotype. These GDF15-activated cells do not project to either the PBN or the ovBNST, but a few do project to the PVH. The others may represent local interneurons or potentially be responding to descending pathways.

## Blocking CCK signalling attenuates the anorexia caused by GDF15

To confirm the importance of GFRAL/CCK[AP/NTS] neurons in mediating the anorectic effects of GDF15, we used a recombinant adeno-associated virus (AAV) expressing a Cre-dependent designer pro-caspase and its activator, the Tobacco Etch Protease (flex-taCasp3-TEVp) to commit CCK[AP/NTS] neurons to cell-autonomous apoptosis. *Cck*[Cre]::eYFP mice were injected into the AP/NTS with the AAV-caspase or an AAV expressing mCherry to control for viral load and transduction efficiency. Post hoc examination confirmed that the designer caspase achieved an effective ablation of CCK[AP/NTS] neurons, assessed by immunostaining for eYFP and also GFRAL (*Figure 4A* and *Figure 4—*

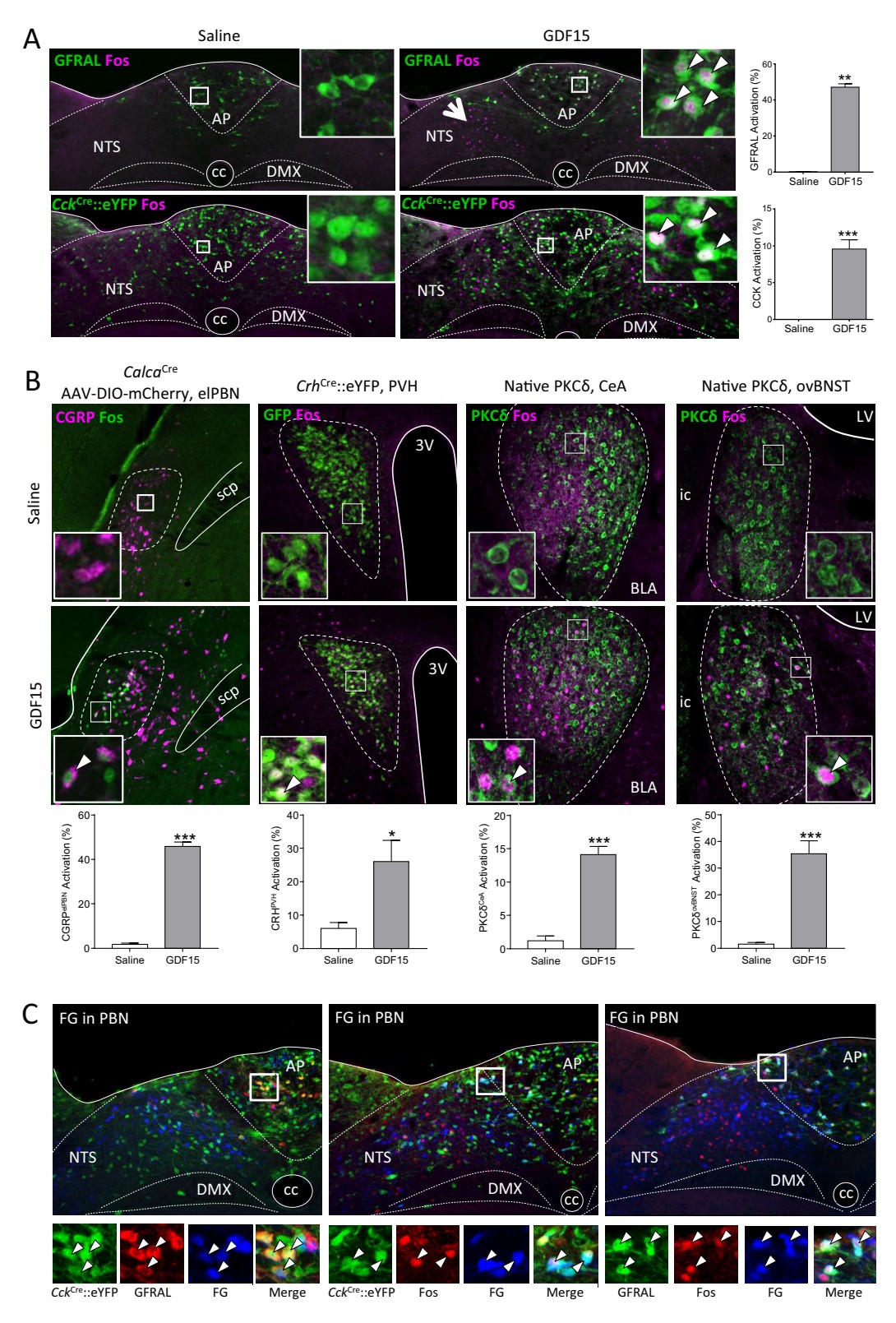

**Figure 3.** GDF15 activates GFRAL+ve/CCK neurons in the AP/NTS. (**A**) Fluorescence photomicrographs showing Fos expression (magenta) in GFRAL and CCK-positive neurons (green) in the AP/NTS following a minimal effective dose of GDF15. For triple labelling, see *Figure 3—figure supplement 1A*. The percentage of activated GFRAL-immunopositive or *Cck*Cre::eYFP neurons is presented on the right (n = 6–7 per group). White arrows in higher magnification insets indicate co-labelled cells. Note that GDF15 administration activated a group of cells in the medial NTS which are GFRAL-ve and

*Figure 3 continued on next page*

*Figure 3 continued*

CCK-ve (large arrow head). (**B**) Dual-label immunofluorescence for Fos and downstream neuronal targets. CGRP neurons were visualised by injecting *Calca*^Cre mice with AAV-DIO-mCherry (Fos green; CGRP magenta). In the other pictures, PKC-δ^+ or *Crh*^Cre::eYFP cells are coloured green. Quantification is provided below the relevant photomicrographs (*Calca*^Cren = 4 per group; *Crh*^Cren = 6–7 per group; PKC-δ^+n = 6–7 per group). (**C**) GFRAL neurons, which were activated by GDF15, project directly to the parabrachial nucleus, as demonstrated using Fluoro-Gold retrotracing. White arrows in higher magnification insets indicate triple-labelled cells. aca (anterior part of the anterior commissure), AP (area postrema), BLA (basolateral amygdala), ovBNST (bed nucleus of the stria terminalis, oval sub-nucleus), cc (central canal), CeA (central nucleus of the amygdala), DMX (dorsal motor nucleus of the tenth cranial nerve, vagus), ic (internal capsule), LV (lateral ventricle), NTS (nucleus of the tractus solitarius), PBN (parabrachial nucleus), PVH (paraventricular nucleus of the hypothalamus), scp (superior cerebellar peduncle), 3V (third ventricle). *p<0.05, **p<0.01, ***p<0.001; unpaired t-test.

The online version of this article includes the following figure supplement(s) for figure 3:

**Figure supplement 1.** Neuronal activation by GDF15.

---

*figure supplement 1*). Over the 10-week period following viral injection, there was no significant difference in body weight between groups (*Figure 4—figure supplement 1*). However, whereas mice transduced with control AAV-mCherry responded to GDF15 with a significant decrease in night-time food intake, those bearing cell-specific ablation of CCK^AP/NTS neurons showed an abrogated response (*Figure 4B*). We further injected GDF15 into mice which had been pre-treated with the CCK receptor antagonist, devazepide. Compared with mice receiving a vehicle control injection, those receiving devazepide displayed an attenuated anorectic response to a subsequent single injection of GDF15 (*Figure 4C*). At early time points, the anorectic effect of GDF15 was reduced by approximately half.

## GFRAL receptor blockade with a monoclonal antibody alleviates the adverse side effects of the cancer therapeutic drug, cisplatin

Finally, the platinum-based therapeutic drug, cisplatin, causes a long-term reduction in food intake, which can have a major contribution to mortality in cancer patients treated with the drug (*Dasari and Tchounwou, 2014*). There is recent evidence that cisplatin acts through the NTS → CGRP^PBN axis in both rats (*Alhadeff et al., 2015*; *Alhadeff et al., 2017*) and mice, (*Hsu et al., 2017*) and the latter study indicates that GFRAL knock-out mice are protected against the anorexic effects of cisplatin. In our hands, a single injection of cisplatin (4 mg/kg, i.p.) at the beginning of the dark phase, led to a reduction in both food intake and body weight which lasted for three days (*Figure 5A and B*). In the same animals, there was a significant increase in circulating GDF15, which also lasted for between 2 and 3 days (*Figure 5C*). After 1 day, GDF15 levels were 45 pg/ml and 270 pg/ml in vehicle- and cisplatin-treated mice, respectively. Food intake and weight loss at 2 days correlated directly with the plasma level of GDF15 (*Figure 5—figure supplement 1*). In a separate experiment, we found also that cisplatin at the same dose activated GFRAL^AP/NTS neurons, as assessed by Fos immunoreactivity (*Figure 5D*).

We have shown previously that weight loss induced in rats by a long-lasting recombinant GDF15 protein can be blocked by the single, subcutaneous administration of a selective monoclonal antibody raised against its receptor (GFRAL mAb) (*Emmerson et al., 2017*). Further, here we show that the reduction in food intake and body weight in mice, caused by three daily doses of native GDF15, is blocked by pre-administration of 10 mg/kg of the GFRAL mAb (*Figure 5E and F*). Importantly, this high dose of GFRAL mAb which effectively blocks GDF15 signalling, did not affect either food intake or body weight when injected alone, signifying that GFRAL is unlikely to have a role in either satiety signalling or normal energy balance (for full mAb dose-response data see *Figure 5—figure supplement 1*). Having determined an effective dose of the GFRAL mAb in mice, this was then used to investigate the role of GDF15/GFRAL signalling in mediating the effects of cisplatin. The GFRAL mAb was administered 1 day before cisplatin and, as before, by itself did not affect either food intake (*Figure 5G*) or body weight (*Figure 5H*). However, pre-administration of the GFRAL mAb completely blocked the anorectic action of cisplatin (*Figure 5G*) and prevented the cisplatin-induced body weight loss (*Figure 5H*).

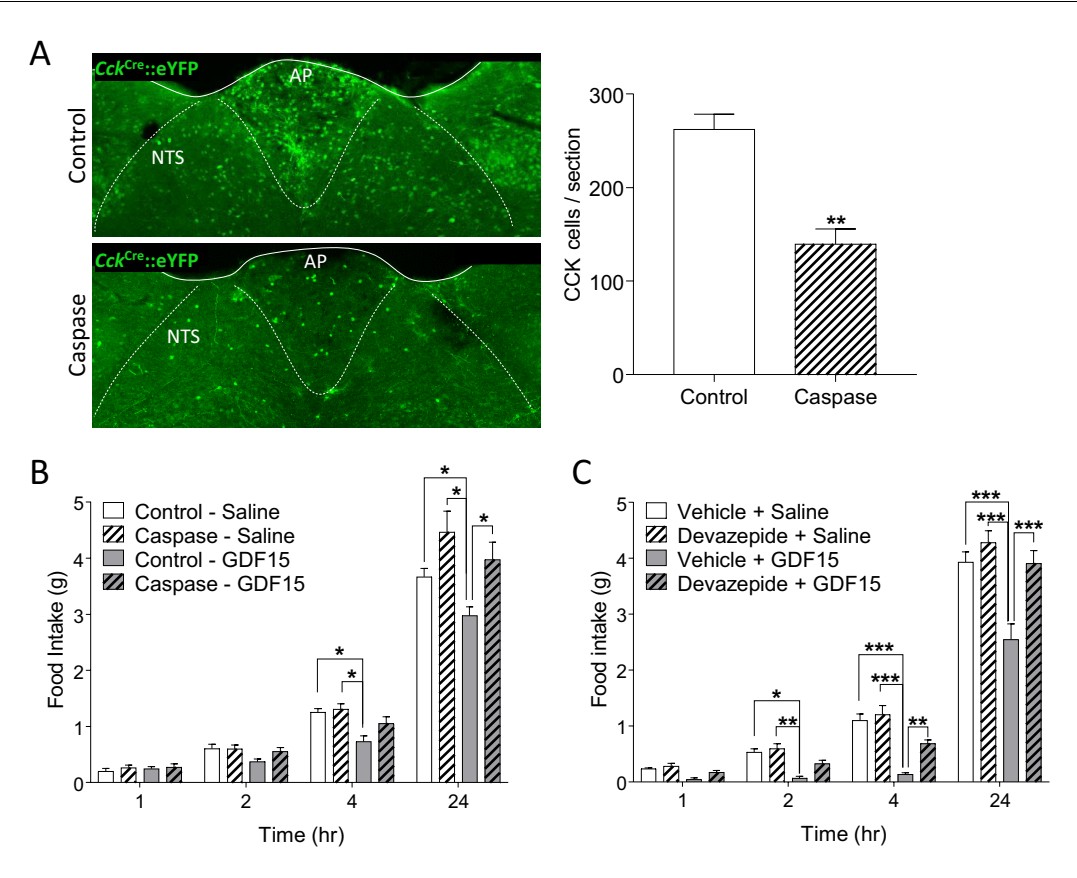

**Figure 4.** GDF15-induced anorexia is dependent on CCK signalling. (**A**) Injection of AAV-caspase into the AP and dorsal NTS of *Cck*^Cre^::eYFP mice caused a reduction in the number of eYFP cells as assessed by immunohistochemistry (n = 7 per group; **p<0.01, unpaired t-test). (**B**) *Cck*^Cre^::eYFP mice transduced with control AAV displayed a significant decrease in food intake following GDF15 administration, while those transduced with AAV-caspase showed reduced anorexia (n = 7 per group; *p<0.05; two-way ANOVA, followed by a post hoc Tukey test). (**C**) Pre-administration of the CCK receptor antagonist, devazepide, attenuated the anorectic response to GDF15 (n = 6 per group; *p<0.05, **p<0.01, ***p<0.001; two-way ANOVA, followed by a post hoc Tukey test).

The online version of this article includes the following figure supplement(s) for figure 4:

**Figure supplement 1.** Treatment with caspase leads to a significant loss of GFRAL neurons.

## Discussion

The recent identification of the receptor for GDF15 and its localisation to a small population of cells in the dorsomedial medulla oblongata, an area involved in gut-brain signalling, ignited interest in the cytokine as having a potential role in body-weight regulation (*Mullican et al., 2017*; *Yang et al., 2017*; *Emmerson et al., 2017*; *Hsu et al., 2017*). Over expression of GDF15 leads to large reductions in both food intake and body weight (*Tsai et al., 2018*; *Johnen et al., 2007*; *Macia et al., 2012*; *Chrysovergis et al., 2014*; *Xiong et al., 2017*), whilst the obese phenotype reported for both the GDF15 and GFRAL knock-out mice is supporting evidence for a homeostatic role in normal body-weight regulation and, perhaps, satiety signalling (*Tsai et al., 2013*; *Low et al., 2017*; *Tran et al., 2018*; *Mullican et al., 2017*; *Hsu et al., 2017*). Careful examination of these null animals also provides results which would not be expected if this were the case. Thus, increases in food intake are either sex specific or only clearly apparent in mice fed high-energy diet. However, one of the more obvious phenotypes is a significant reduction in overall locomotor activity, which might explain a major part of the obesity (*Tsai et al., 2013*; *Tran et al., 2018*). Furthermore, the measurement of plasma levels of GDF15 in humans or mice does not correlate with meal times or nutritional status, which again would argue against it being a circulating satiety factor (*Patel et al., 2019*; *Tsai et al., 2015*). By contrast, GDF15 levels are greatly enhanced in a number of disease states,

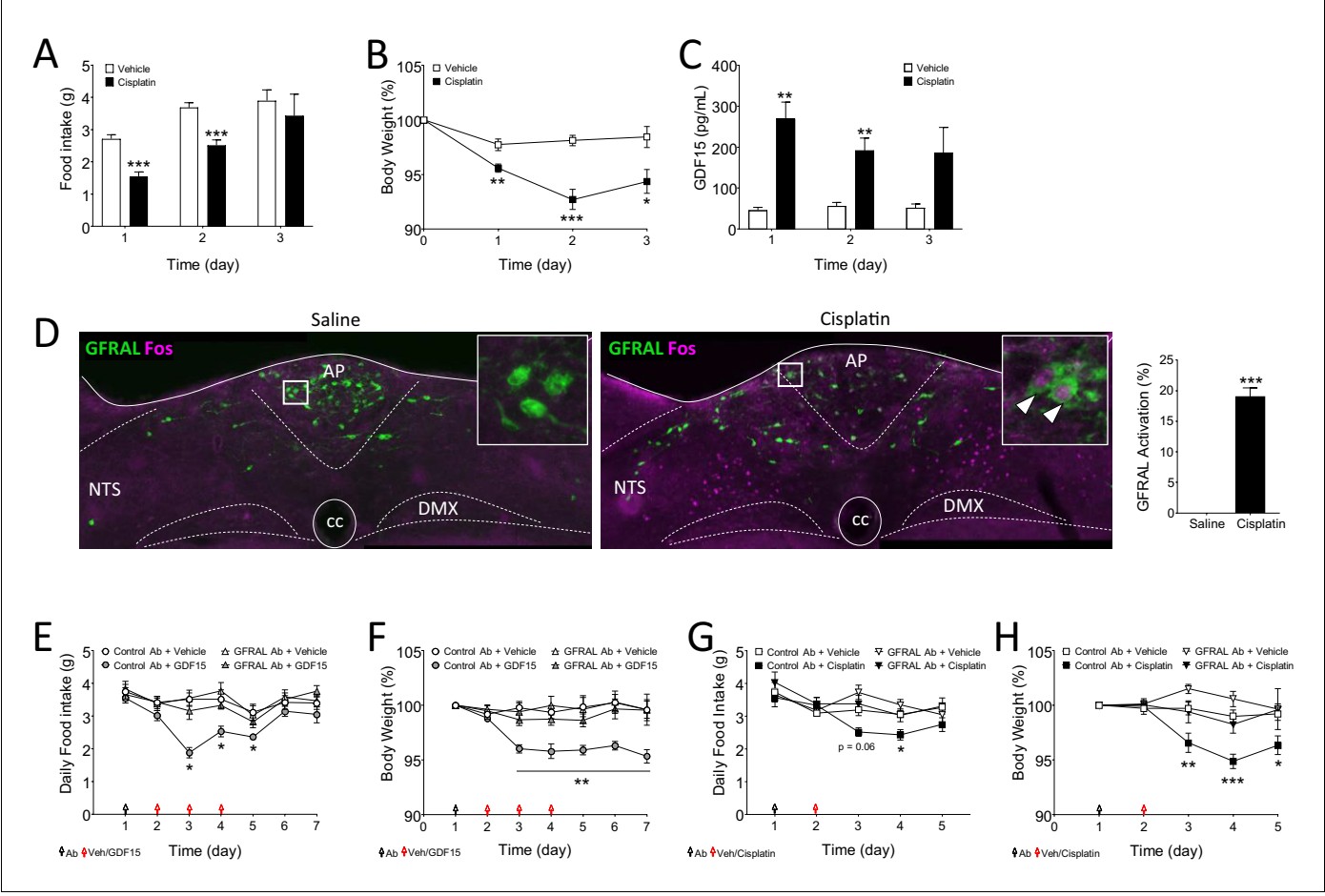

**Figure 5.** The anorectic action of the cancer therapeutic drug, cisplatin, is blocked by inhibition of signalling through GFRAL. (**A**) A single dose of cisplatin reduced food intake and (**B**) body weight over the following 3 days (n = 6 per time point; *p<0.05, **p<0.01, ***p<0.001; two-way ANOVA, followed by a post hoc Tukey test). (**C**) This corresponded with an increase in circulating GDF15 (n = 6 per time point, ***p<0.001, unpaired t-test) and (**D**) induction of Fos (magenta) in immunopositive GFRAL neurons (green) at 24 hr after administration (n = 5–6 per group; ***p<0.001, unpaired t-test). (**E**) Three injections of GDF15, on days 2–4, led to a decrease in cumulative food intake and (**F**) body weight (n = 6 per group; *p<0.05, **p<0.01; repeated measures ANOVA, followed by a post hoc Tukey test, control Ab + GDF15 versus all other groups). The actions of GDF15 were blocked completely by pre-administration of a monoclonal antibody against GFRAL (10 mg/kg) on day 1. The GFRAL mAb had no effect on food intake or body weight by itself. For full data set, using different concentrations of GFRAL mAb, see *Figure 5—figure supplement 1B and C*. (**G**) Pre-administration of 10 mg/kg GFRAL mAb the day before, completely blocked the reduction of food intake and (**H**) body weight caused by cisplatin (n = 5–6 per group; *p<0.05, **p<0.01, ***p<0.001; repeated measures ANOVA, followed by a post hoc Tukey test).

The online version of this article includes the following figure supplement(s) for figure 5:

**Figure supplement 1.** The anorectic action of the cancer therapeutic drug, cisplatin, is blocked by inhibition of signalling through GFRAL.

including cancer, cardiovascular disease and chronic kidney disease, pointing towards a more obvious role as a non-physiological anorectic signal (*Patel et al., 2019*; *Welsh et al., 2003*; *Kempf et al., 2007*; *Ho et al., 2013*; *Bauskin et al., 2006*; *Luan et al., 2019*). In fact, the only non-chronic disease situation described, in which GDF15 is up regulated, is *hyperemesis gravidarum* associated with intense nausea and vomiting in early pregnancy (*Fejzo et al., 2018*; *Petry et al., 2018*). Thus, while we will not rule out a physiological role for GDF15, most of the literature seems to point towards it having a more significant role in sickness behaviour. This is strongly supported by our results that show that systemic administration of GDF15 causes robust conditioned taste

aversion, conditioned place aversion and pica behaviour, and that blocking endogenous GFRAL signalling with a selective monoclonal antibody does not affect baseline food intake or body weight.

GFRAL-positive neurons span the AP and the inner border with the NTS, a region synonymous with responses to toxins and other nausea-inducing agents, and which lacks a blood-brain barrier (*Miller and Leslie, 1994*). The AP and NTS are also part of the dorsal vagal complex, which responds to gut-brain signalling. GFRAL is not expressed in the gut or in the vagus nerve, and since GDF15 is still effective at inducing anorexia in vagotomised rats, (*Yang et al., 2017*) the simplest conclusion is that circulating GDF15 acts directly on GFRAL expressed in the AP/NTS. We and others have argued for the existence of specific neuron types in this region of the brainstem which respond selectively to different sensory modalities (for example satiety versus nausea) in order to reduce food intake (*Luckman and Lawrence, 2003*; *Kreisler et al., 2014*). Thus, it is important to identify the phenotype of GFRAL cells. Although evidence has been provided that GFRAL is expressed in TH-positive catecholaminergic neurons, until now this has not been accurately quantified (*Yang et al., 2017*; *Tsai et al., 2014*). Here, using a variety of models, we conclude that the major, identifiable population of GFRAL neurons contain the neuropeptidergic transmitter, CCK. GFRAL-positive neurons respond to GDF15, but not with many other stimuli, and the action of GDF15 can be abrogated either by genetically ablating CCK neurons selectively in this region or by blocking CCK transmission. Thus, we propose that the primary target for GDF15 is a distinct population of GFRAL/CCK neurons which span the AP/NTS to engage well-characterised circuitry involved in anorexia and conditioned aversion (*Wu et al., 2012*; *Carter et al., 2013*; *Chen et al., 2018*). We have found that CCK neurons activated by GDF15 project directly to the PBN, and that other downstream targets, in the PVH, CeA and ovBNST are also involved. There is ample evidence for anorectic signals to utilise parallel downstream pathways, but with convergence at specific nodes which we demonstrate are activated by GDF15, including CGRP neurons in the PBN (*D'Agostino et al., 2016*; *Roman et al., 2016*; *Roman et al., 2017*) and/or PKC-$\delta^+$ neurons in both the CeA and ovBNST (*Cai et al., 2014*; *Wang et al., 2019*). There is still much to learn about these pathways, not least because we have shown that GFRAL neurons are not activated by the archetypal nausea-inducing agent LiCl, nor does GDF15 activate PPG (here) or GLP-1 receptor neurons (*Welsh et al., 2003*; *Frikke-Schmidt et al., 2019*), both capable of transmitting nauseous signals. Recently, bacterial or viral infections have been associated with secretion of GDF15 and an adaptive response in order to increase pathogen tolerance (*Luan et al., 2019*). In this ground-breaking paper, it was demonstrated that GDF15/GFRAL signalling increases triglyceride production by the liver in order to protect tissues, which are dependent on triglycerides for fuel, from metabolic damage due to inflammation. Also, the loss of appetite and weight caused by metformin in diet-induced obese mice is dependent on GDF15/GFRAL signalling, as is the accompanying increase in insulin sensitivity (*Coll et al., 2020*).

It is interesting to note that increased GDF15 has been measured in the circulation of human cancer subjects (*Welsh et al., 2003*; *Bauskin et al., 2006*; *Brown et al., 2003*) and, recently an NTS → CGRP$^{PBN}$ → CeA/ovBNST axis has been implicated in mediating the anorexia associated with cancer models in mice (*Campos et al., 2017*). Likewise, work in rats has demonstrated that this pathway appears also to be activated by the platinum-based, cancer therapeutic drug, cisplatin (*Alhadeff et al., 2015*; *Alhadeff et al., 2017*). Although in neither case has the primary brainstem neuron been identified (*Hsu et al., 2017*), it is reported that weight loss caused by cisplatin is reduced in GFRAL knock-out mice, and we now show that this is true also if wild-type mice are pretreated with a neutralising GFRAL antibody. Although it is yet to be verified, the possibility exists that both a disease state (cancer) and the treatment (cisplatin) may exacerbate anorexia through the same brainstem pathway. If this is the case, then either GFRAL neutralising antibodies or GFRAL antagonists may provide a possible co-treatment opportunity for patients suffering with cancer-related anorexia/cachexia. The caveat to this is that the secretion of GDF15 during cancer, as it appears so for inflammatory infections, is presumably an adaptive response, so blocking GDF15/GFRAL signalling may worsen the disease and other symptoms. GDF15 has been located in different tissues when they become cancerous (*Welsh et al., 2003*; *Buckhaults et al., 2003*). If GDF15 has an adaptive systemic effect, then it may be possible to bypass this and, instead, selectively target the central pathways downstream of brainstem GFRAL.

# Materials and methods

## Key resources table

| Reagent type (species) or resource | Designation | Source or reference | Identifiers | Additional information |
|---|---|---|---|---|
| Genetic reagent (*Mus musculus*) | C57Bl/6J (Mouse, male) | Envigo | Stock #057 | MGI:2164189 |
| Genetic reagent (*Mus musculus*) | C57Bl/6NHsd (Mouse, male) | Envigo | Stock #044 | MGI:2161078 |
| Genetic reagent (*M. musculus*) | C57Bl/6J (Mouse, male) | Charles River | Stock #632 | MGI:3028467 |
| Genetic reagent (*M. musculus*) | C57Bl/6J (Mouse, male) | Janvier labs | N/A | MGI:2670020 |
| Genetic reagent (*M. musculus*) | $Pomc^{eGFP}$ (Mouse, male) | Jackson Laboratories | Stock #: 009593 | MGI:3851684 |
| Genetic reagent (*M. musculus*) | $Cck^{ires\text{-}Cre}$ (Mouse, male) | Jackson Laboratories | Stock #: 012706 | MGI:5014249 |
| Genetic reagent (*M. musculus*) | $Crh^{ires\text{-}Cre}$ (Mouse, male) | Jackson Laboratories | Stock #: 012704 | MGI:4452101 |
| Genetic reagent (*M. musculus*) | $Slc17a6^{ires\text{-}Cre}$ (Mouse, male) | Jackson Laboratories | Stock #: 016963 | MGI:5300532 |
| Genetic reagent (*M. musculus*) | Rosa26-loxSTOPlox-eYFP (Mouse, male) | Jackson Laboratories | Stock #: 006148 | MGI:3621481 |
| Genetic reagent (*M. musculus*) | $Gcg^{iCre}$ (Mouse, male) | *Parker et al., 2012* PMID:22638549 | N/A | MGI:5432481 |
| Genetic reagent (*M. musculus*) | $Prlh^{ires\text{-}Cre}$ (Mouse, male) | *Dodd et al., 2014* PMID:25176149 | N/A | MGI:5634277 |
| Genetic reagent (*M. musculus*) | $Calca^{Cre}$ (Mouse, male) | *Carter et al., 2013* PMID:24121436 | N/A | MGI:5559692 |
| Genetic reagent (Rattus norvegicus) | Sprague Dawley (Rat, male) | Envigo | Stock #: SD-002 | N/A |
| Antibody | anti-cFos (Rabbit polyclonal) | Santa Cruz | Cat.# SC52 RRID:AB_2106783 | Primary antibody (1:500) IHC |
| Antibody | anti-DS Red (Goat polyclonal) | Santa Cruz | Cat.# 33353 RRID:AB_639924 | Primary antibody (1:500) IHC |
| Antibody | anti-GFP (Chicken polyclonal) | Abcam | Cat.# 13970 RRID:AB_300798 | Primary antibody (1:2000) IHC |
| Antibody | anti-GFRAL (Sheep polyclonal) | Thermofisher | Cat.# PA5-47769 RRID:AB_2607220 | Primary antibody (1:200) IHC |
| Antibody | anti-GLP1 (Rabbit polyclonal) | PenLabs | Cat.#. T-4363 RRID:AB_518978 | Primary antibody (1:2000) IHC |
| Antibody | anti-PKCδ (Mouse monoclonal) | BD Biosciences | Cat.#. 610398 RRID:AB_397781 | Primary antibody (1:500) IHC |
| Antibody | anti-TH (Rabbit polyclonal) | AbCam | Cat.# AB112 RRID:AB_297840 | Primary antibody (1:2000) IHC |
| Antibody | anti-TH (Sheep polyclonal) | Millipore | Cat.# AB1542 RRID:AB_90755 | Primary antibody (1:1000) IHC |
| Antibody | anti-chicken, Alexa Fluor 488 (Donkey polyclonal) | Jackson Immuno Research | Cat.# 703-545-155 RRID:AB_2340375 | Secondary antibody (1:1000) IHC |
| Antibody | anti-mouse, Alexa Fluor 594 (Donkey polyclonal) | Jackson Immuno Research | Cat.# 715-585-150 RRID:AB_2340854 | Secondary antibody (1:1000) IHC |
| Antibody | anti-rabbit, Alexa Fluor 350 (Donkey polyclonal) | Molecular Probes | Cat.# A10039 RRID:AB_2534015 | Secondary antibody (1:1000) IHC |
| Antibody | anti-Sheep, Alexa Fluor 350 (Donkey polyclonal) | Molecular Probes | Cat.# A21097 RRID:AB_10376162 | Secondary antibody (1:1000) IHC |

*Continued on next page*

*Continued*

| Reagent type (species) or resource | Designation | Source or reference | Identifiers | Additional information |
|---|---|---|---|---|
| Antibody | anti-sheep, Alexa Fluor 594 (Donkey polyclonal) | Molecular Probes | Cat.# A11016 RRID:AB_10562537 | Secondary antibody (1:1000) IHC |
| Antibody | Anti-GFRAL (Mouse monoclonal) | *Emmerson et al., 2017* PMID:28846098 | mIgG1 GFRAL 8A2 | Subcutaneous injection (0–10 mg/kg) |
| Recombinant DNA reagent | AAV8-hSyn-DIO-mCherry | Dr Bryan Roth Addgene | Cat.# 50459-AAV8 | N/A |
| Recombinant DNA reagent | AAV5-flex-taCasp3-TEVp | Dr Nirao Shah University of North Carolina Vector Core | N/A | PMID:23663785 |
| Sequence-based reagent | *Gfral* | Advanced Cell Diagnostics | Cat.# 417021-C3 | RNAScope mRNA probe |
| Sequence-based reagent | *Cck* | Advanced Cell Diagnostics | Cat.# 402271-C1 | RNAScope mRNA probe |
| Sequence-based reagent | *Cckr1* | Advanced Cell Diagnostics | Cat.# 313751-C1 | RNAScope mRNA probe |
| Sequence-based reagent | *Calca* | Advanced Cell Diagnostics | Cat.# 420361-C2 | RNAScope mRNA probe |
| Peptide, recombinant protein | GDF15 | R and D Systems | Cat.# 9279-GD | (4 nmol/kg) |
| Peptide, recombinant protein | Streptavadin 488 | Jackson Immuno Research | Cat.# 016-540-084 RRID:AB_2337249 | (1:1000) IHC |
| Peptide, recombinant protein | Streptavadin 594 | Jackson Immuno Research | Cat.# 016-580-084 RRID:AB_2337250 | (1:1000) IHC |
| Commercial assay or kit | Mouse/rat GDF15 ELISA | R and D Systems | Cat.# MGD-150 | N/A |
| Commercial assay or kit | RNAscope Multiplex Fluorescent Assay | Advanced Cell Diagnostics | Cat # 323100 | |
| Chemical compound, drug | Devazepide | Tocris Bioscience | Cat.# 2304 | (1 mg/kg) |
| Chemical compound, drug | Lithium chloride (LiCl) | Sigma | Cat L9650 (mouse) Cat.# 73036 (rat) | Mouse (128 mg/kg) Rat (128 mg/kg) |
| Chemical compound, drug | Hydroxystibamidine (Fluoro-Gold) | Invitrogen, Thermofisher | Cat.# H22845 | 4% in $H_2O$ |
| Chemical compound, drug | Cisplatin | Sigma Aldrich | Cat.# PHR1624 | (4 mg/kg) |
| Software, algorithm | Prism | GraphPad | RRID:SCR_002798 | Version 7 |
| Software, algorithm | Fiji | ImageJ | RRID:SCR_002285 | Version 2.0.0-rc-69/1.52 p |
| Software, algorithm | Micromanager | ImageJ | RRID:SCR_016865 | Version 1.4.23 |
| Software, algorithm | Smart | Panlab, Harvard Biosciences/Biochrom Ltd | RRID:SCR_002852 | Version 3.0 |

## Animals

Non-transgenic C57Bl/6 mice were obtained from Charles River (Manston, Kent, UK), Envigo (Huntingdon, UK and Indianapolis IN) or Janvier Labs (Le Genest-Saint-Isle, France). *Pomc*[eGFP], *Cck*[ires-Cre], *Crh*[ires-Cre] (*Taniguchi et al., 2011*), *Slc17a6*[ires-Cre] and Rosa26-loxSTOPlox-eYFP were all purchased from Jackson Laboratories (stock numbers 009593, 012706, 012704, 016963 and 006148, respectively; Bar Harbor, ME). We have described the generation of the PPG (*Gcg*[Cre]) and PrRP (*Prlh*[ires-Cre]) mice (*Parker et al., 2012*; *Dodd et al., 2014*). *Calca*[ires-Cre] mice (*Carter et al., 2013*) were a kind gift from Prof Richard Palmiter (Howard Hughes Medical Institute, University of Washington).

## Drugs and viruses

Recombinant human GDF15 was either made in house as described (*Emmerson et al., 2017*) or purchased from R and D Systems (Abingdon, UK). GDF15 was initially dissolved in 15 mM HCl, neutralised and diluted in saline. Devazepide was purchased from Tocris Bioscience (Bristol, UK) and dissolved in 40% DMSO. Cisplatin (Sigma-Aldrich, Gillingham, UK) was dissolved directly in saline.

Fluoro-Gold (hydroxystilbamidine, 4% w/v solution in water; Invitrogen, ThermoFisher, MA) was injected into mice anaesthetised with isoflurane (2–3% in oxygen) and placed in a stereotaxic frame. The skull was exposed and holes drilled at the site of injection. Fluoro-Gold was delivered unilaterally via a glass micropipette affixed to a Nanoject II Auto Nanoliter Injector (Drummond Scientific Company, PA) using co-ordinates as determined in the Mouse Brain Atlas: (*Paxinos and Franklin, 2004*) elPBN, −4.9 mm A/P, −1.4 mm M/L, −3.8 mm D/V from *bregma* (12 nl); ovBNST, +0.3 mm A/P, −1.0 mm M/L, −4.5 mm D/V (18 nl); PVH, - 0.7 mm A/P; −0.3 mm M/L; −5.5 mm D/V (18 nl). All animals were left to recover for 2 weeks to allow axonal transport before being transcardially perfused (see below).

Viral injections into the AP/NTS were performed as described previously with minor modifications (*D'Agostino et al., 2016*; *D'Agostino et al., 2018*). Briefly, 9- to 11-week-old male mice were anaesthetised with a mixture of ketamine and xylazine dissolved in saline (80 and 10 mg/kg, respectively; 10 ml/kg i.p.). Mice were placed in a stereotaxic frame, an incision was made at the level of the *cisterna magna,* and neck muscles were carefully retracted. Following *dura* incision, the *obex* served as reference point for injections with a glass micropipette. AP/NTS coordinates were approximatively 0.2 mm A/P, 0 and ±0.2 mm M/L, −0.2 mm D/V from *obex*. About 150 nl of virus were delivered during each of the three microinjections. Animals were administered analgesia (5 mg/kg Carprofen, s.c.) for 2 days post-operatively and given a minimum of 14 days recovery before night-time feeding measurement. AAV5-mCherry and AAV5-flex-taCasp3-TEVp were obtained from Addgene (Watertown, MA) and the University of North Carolina Vector Core (Chapel Hill, NC), respectively.

## Tissue preparation and histology

For all immunohistochemical experiments, animals were deeply anaesthetised with 4% isoflurane in oxygen and transcardially perfused with heparinsed saline (20,000U per litre in 0.9% NaCl) followed by 4% paraformaldehyde in 0.1 M phosphate buffer. Brains were dissected and post-fixed overnight at 4°C and then cryoprotected in 30% sucrose. Brains were cut into 30-μm-thick coronal sections using a freezing sledge microtome (Bright 8000, Cambridge, UK) and either processed immediately or stored in cryoprotectant solution at −20°C.

Immunohistochemistry was performed on free-floating sections at room temperature unless stated otherwise. All antibodies are listed in the Key Resource Table. Brain sections were washed in 0.2% Triton X-100 in 0.1 M phosphate buffer and blocked in 5% normal serum for 1 hr, before being incubated in primary antibody (made up in to 1% normal serum) overnight at 4°C. The next day, sections were washed again and then incubated in secondary antibody for 2 hr. Sections were washed and, where biotinylated secondary antibodies were used, incubated for a further hour in streptavidin-conjugated fluorophores diluted in phosphate buffer. Finally, sections were washed in water, mounted onto glass slides, air-dried overnight and coverslipped with ProLong Gold (Thermo Fisher Scientific, MA). Sections were visualised on a Zeiss Axiomanager.D2 upright microscope (Zeiss, Oberkochen, Germany) and images captured using a Coolsnap HQ1 camera (Photometics, AZ) through Micromanager software v1.4.23 (https://imagej.net/Micro-Manager). Specific band pass filter sets for DAPI, FITC and Texas Red were used to prevent bleed through from one channel to the next. All images were processed and analysed in Fiji ImageJ (https://fiji.sc/).

Eight-week-old C57BL/6J mice (n = 3) were anaesthetised by $CO_2$, decapitated, and the brains removed and snap frozen on crushed dry ice. Four or five 10-μm-thick tissue sections at the level of the AP or PBN were collected for RNA in situ hybridisation histology for *Gfral* (cat#417021-C3) and *Cck* (cat# 402271-C1) or *Cckr1* (cat#313751-C1) and *Calca* (cat#420361-C2), respectively. mRNA was detected using RNAscope Multiplex Fluorescent Assay reagent kits (Advanced Cell Diagnostics, Inc, Newark, CA), according to the manufacturer's instructions, at Gubra (Hørsholm, Denmark). Slides were counter stained with DAPI to identify cellular nuclei. Slides were scanned under a 20X objective in an Olympus VS120 Fluorescent scanner.

## Feeding and body-weight studies

For GDF15 night-time feeding experiments, food was removed from the animals for 2 hr before lights out. At lights out, mice were administered GDF15 subcutaneously and food was returned at the same time. Food intake was recorded at 0, 1, 2, 4, and 24 hr after injection of GDF15. Devazepide (1 mg/kg, i.p) was administered 45 min before GDF15. For the fast-refeeding experiment, food was removed from the mice at lights out on the night before the experiment. After a 16.5 hr fast, mice were administered GDF15 and food returned. Food intake was measured at the same time points (n = 5–7 per group).

C57Bl/6J male mice (aged 10 = 18 weeks; n = 6 for each group) were administered a single, intraperitoneal dose of saline or cisplatin (4 mg/kg). Food intake and body weight were monitored daily. At 24, 48, and 72 hr, n = 6 mice from each group were sacrificed with $CO_2$ inhalation and blood was collected with cardiac puncture. Plasma was taken with Approtinin and DPP4 inhibitors. GDF15 levels are measured using a mouse specific ELISA, according to manufacturer's directions (R and D Systems, MGD150). A dose-finding experiment for GFRAL mAb (mIgG1 GFRAL 8A2, Lilly Indianapolis, USA) was performed. Briefly, 1 day prior to GDF15 administration at 4 nmol/kg for 3 consecutive days, the mice were subcutaneously dosed once either with control antibody (mIgG1 antibody, Lilly Indianapolis) at 10 mg/kg or ascending doses of antibody at 0.3, 1, 3 and 10 mg/kg of GFRAL mAb. Daily food intake and body weight were measured for three days. In a different experiment, control antibody or GFRAL mAb (both 10 mg/kg) were subcutaneously dosed once 1 day prior to intraperitoneal dose of cisplatin at 4 mg/kg.

## Conditioned taste and place aversion tests, pica behaviour

For conditioned taste aversion, C57Bl/6J mice were housed in cages that permitted ad libitum access to water from two bottles, side-by-side, for at least 1 week before the experiment. On day 1 of the study, animals were water deprived overnight for 16.5 hr. The following morning (day 2), water-deprived animals were provided with two bottles of a novel 15% sucrose solution (dissolved in drinking water) for 30 min. At the end of the 30 min sucrose exposure, animals received an s.c. injection of either saline or GDF15 (4 nmol/kg, 4 ml/kg; n = 6 per group). Two water bottles were returned immediately and mice had unlimited access to water for one night. On day 3, mice were again water deprived overnight. On day 4, water-deprived animals were provided with one bottle of 15% sucrose and one bottle of water for a period of 24 hr. Volumes of sucrose and water intake were measured at 2 hr and 24 hr and used to calculate sucrose preference (sucrose intake/total fluid intake * 100). Food was available ad libitum throughout the study. The positioning of the sucrose and water bottles (left or right) was randomised within treatment groups.

Conditioned place aversion was performed using an apparatus composed of two chambers with distinct visual and tactile qualities, connected by a brightly lit corridor (Harvard Biosciences/Biochrom Ltd., Cambridge, UK). The darker chamber consisted of a rough black floor and black spotted walls, whereas the lighter chamber consisted of a smooth grey floor and grey striped walls. Time spent in each chamber was monitored by video cameras mounted directly above the apparatus, connected to a computer running tracking software (Smart v3.0, Panlab, Harvard Biosciences/Biochrom Ltd.). All procedures were carried out between three and five hours after lights on. On day 1, C57Bl/6J mice (n = 12) were given free access to the full apparatus and allowed to freely explore both chambers for 30 min. Their initial pre-test preference was calculated from the time spent in each chamber. A biased design was used, whereby GDF15 was associated with the most-preferred chamber, which was the darker chamber for all mice. On days 2 and 3, a conditioning session was performed, where mice were restricted to one chamber following administration of either GDF15 (4 nmol/kg, injected s.c. on day 2) or saline (0.9% NaCl injected s.c. on day 3). Following each injection, mice were returned to their home cage for 10 min and then placed in the relevant chamber for 30 min. On day 4, a test session was performed in identical fashion to day 1, and their post-conditioning preference was calculated from the time spent in each chamber. Food and water was available ad libitum throughout the study, except for when mice were in the conditioning apparatus.

To measure pica behaviour, male Sprague-Dawley rats (Envigo, Indianapolis, IN) were acclimated with kaolin pellets available ad libitum in a hopper placed continuously in the home cage. Rats were assigned randomly to three groups (n = 9–10 per group) and treated on 3 consecutive days with vehicle (acetate buffer, pH 5.5, 1 ml/kg, s.c.), human recombinant GDF15 (0.2 mg/kg, s.c.) or LiCl

(0.3 M in water, 1% body weight, equivalent to 128 mg/kg, i.p.). Body weight, chow and kaolin intake were determined daily.

## Statistics

Statistical analyses were performed using Prism 7 (Graphpad Software, La Jolla, CA). Data were analysed using t-test, one-way ANOVA, two-way or repeated measures ANOVA with post hoc comparisons. When appropriate, non-parametric equivalents were used. $N$ represents independent biological replicates. No statistical methods were used to predetermine sample sizes. Sample size was computed based on pilot data and published literature. Data are presented as mean ± SEM and statistical significance was set at $p < 0.05$.

# Acknowledgements

This work was funded through BBSRC and MRC grants to SML (BB/M001067/1; BB/L021129/1; BB/S008098/1; MR/R002991/1). RS was funded for part of this project by the award of a University of Manchester PhD Scholarship. GD'A is funded by an MRC Career Development Award (MR/P009824/2).

# Additional information

## Competing interests

Emily C Beebe, James D Dunbar, Jesline T Alexander-Chacko, Dana K Sindelar, Tamer Coskun, Paul J Emmerson: Paid employee of Eli Lilly. Simon M Luckman: BB/S008098/1 is a BBSRC Industrial Partnership Award between SML and Eli Lilly. The other authors declare that no competing interests exist.

## Funding

| Funder | Grant reference number | Author |
| --- | --- | --- |
| Biotechnology and Biological Sciences Research Council | BB/M001067/1 | Simon M Luckman |
| Biotechnology and Biological Sciences Research Council | BB/L021129/1 | Simon M Luckman |
| Medical Research Council | MR/R002991/1 | Simon M Luckman |
| Medical Research Council | MR/P009824/2 | Giuseppe D'Agostino |
| Biotechnology and Biological Sciences Research Council | BB/S008098/1 | Simon M Luckman |
| University of Manchester | PhD Scholarship | Rosemary Shoop |

The funders had no role in study design, data collection and interpretation, or the decision to submit the work for publication.

## Author contributions

Amy A Worth, Conceptualization, Data curation, Formal analysis, Investigation, Visualization, Methodology, Writing - original draft, Writing - review and editing; Rosemary Shoop, Data curation, Formal analysis, Investigation, Visualization; Katie Tye, Emily C Beebe, James D Dunbar, Jesline T Alexander-Chacko, Dana K Sindelar, Formal analysis, Investigation; Claire H Feetham, Investigation; Giuseppe D'Agostino, Data curation, Formal analysis, Investigation, Methodology, Writing - review and editing; Garron T Dodd, Resources, Formal analysis, Investigation; Frank Reimann, Fiona M Gribble, Resources; Tamer Coskun, Paul J Emmerson, Resources, Formal analysis, Investigation, Methodology, Writing - review and editing; Simon M Luckman, Conceptualization, Resources, Supervision, Funding acquisition, Methodology, Writing - original draft, Project administration, Writing - review and editing

## Author ORCIDs

Amy A Worth (iD) https://orcid.org/0000-0003-2573-7140
Rosemary Shoop (iD) https://orcid.org/0000-0002-3617-4358
Simon M Luckman (iD) https://orcid.org/0000-0001-5318-5473

## Ethics

Animal experimentation: All procedures were conducted in accordance with either: the United Kingdom Animals (Scientific Procedures) Act, 1986 (ASPA) and approved by the local animal welfare ethical review body (AWERB); the Eli Lilly Institutional Animal Care and Use Committee (IACUC) in accordance with the National Institutes of Health Guide for Care and Use of Laboratory Animals; or the University of Melbourne Animal Ethics Committee (1914919) and conformed to National Health & 8 Medical Research Council (Australia) guidelines regarding the care and use of experimental animals. Additional guidance from the UK National Centre for 3R's (NC3Rs) was followed where applicable.

## Decision letter and Author response

Decision letter https://doi.org/10.7554/eLife.55164.sa1
Author response https://doi.org/10.7554/eLife.55164.sa2

# Additional files

## Supplementary files

• Supplementary file 1. A summary table of the quantification of cell counts demonstrating neuropeptide co-expression in the medulla oblongata. The number of cells per section single-, double- or triple-labelled on sections through the AP and NTS. Values are stated as are mean ± SEM (n = number of animals). Percentage co-expression is written in the text. Methods involved either immunohistochemistry (top) or in situ hybridisation histology (bottom).

• Transparent reporting form

## Data availability

All data generated or analysed during this study are included in the manuscript and supporting files.

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
