## [Decision Letter]

**Acceptance summary:**

Anorexia is a common response to illness or during cancer treatment and it can have serious consequences. The authors of this paper reveal that the cytokine GDF-15 is produced in response to the cancer drug, cisplatin and that it induces anorexia by activating cells in the hindbrain that express cholecystokinin and the GDF-15 receptor. Blockade of GDF-15 receptor or CCK signaling ameliorates the anorexic effect of cisplatin.

**Decision letter after peer review:**

Thank you for submitting your article "The cytokine GDF15 signals through a population of brainstem cholecystokinin neurons to mediate anorectic signalling" for consideration by *eLife*. Your article has been reviewed by three peer reviewers, including Richard D Palmiter as the Reviewing Editor and Reviewer #1, and the evaluation has been overseen by Catherine Dulac as the Senior Editor. The following individual involved in review of your submission has agreed to reveal their identity: Marcelo O Dietrich (Reviewer #3).

The reviewers have discussed the reviews with one another and the Reviewing Editor has drafted this decision to help you prepare a revised submission.

The role of GDF15 and its receptor GFRAL in signaling peripheral threats is an exciting advance in the field of interoceptive signaling. Understanding how the GFRAL-expressing neurons in the AP/NTS signal to other brain regions is important. Characterizing the signaling molecules made by GFRAL neurons and their relevant post-synaptic targets is critical to this endeavor.

The reviewers are pleased with the overall thrust of the experiments described in this paper. The manuscript includes some important observations, but many details are missing (see below). Consequently, we are not sure whether your team can address these issues in a timely manner (2 months). If not, you may prefer to withdraw the paper now and submit it as a new paper when you have more definitive data.

The major concerns that need to be addressed are as follows:

1) The molecular identity of only about half of the GFRAL neurons was determined. What is the molecular identity of the other half? What fast transmitters do GFRAL neurons use, if any? Single-cell RNA-Seq is one way to answer these questions but is probably not feasible in 2 months unless the authors have already initiated this approach.

2) Addressing the identity of GFRAL post-synaptic neurons that mediate anorexia or other phenotypes is one of the goals of this study. GDF15 activates Fos in parabrachial CGRP neurons, in agreement with previous results, suggesting that GFRAL (CCK subtype) neurons may directly innervate CGRP neurons; however, experiments should be included that reveal whether the signaling is direct or indirect. Is the projection to the PBN the only one that mediates anorexia in response to GFRAL neuron activation?

3) The observation that CCK neurons and CCK itself are an important contributor to the anorexic phenotype is an interesting observation. If the GFRAL/CCK neurons are glutamatergic, it is surprising that CCK is so important. Experiments showing the CCK action in the PBN is important and determining which CCK receptors are involved are needed to solidify this interesting observation.

4) Does CCK ablation in AP/NTS have long-term effects on food intake or body weight. Perhaps this experiment has already been published. If so, provide a reference.

5) The conditioned place aversion experiment needs clarification. The authors say that testing was the same as day one, i.e. 30-min (1800-s) session. But Figure 2D shows only ~500 s on the preferred side which is less than half of the total time. Pairing GDF15 with the preferred side decreased that time to about 250 seconds suggesting that it is aversive. One problem is that "CPA score" is never defined.

6) The number of mice used for each experiment should be included in the figure legends.

The reviewers also have specific suggestions for improvement of this paper which can be found in the individual comments.

Reviewer #1:

The authors of this paper provide data demonstrating that GDF15 induces anorexia and nausea-like behavior in mice/rats and suggest that GDF15 is not a physiological regulator of satiety but rather a sickness signal. They also show that cisplatin stimulates GDF15 expression and the anorexia effect of this cancer-treatment drug depends on GFRAL. Many of the experiments are similar to those recently published by some of the same authors in Cell Metabolism (2020, 31:1-12).

1) The strength of this paper is that the authors begin to characterize the signaling molecules expressed by GFRAL neurons and the neural circuits that they activate. Using a candidate-gene approach they provide evidence that about half of the GFRAL neurons express CCK and/or TH, but the remaining neurons remain unidentified. What fraction of the GFRAL neurons are glutamatergic? Using RNA-Seq of GFRAL-expressing neurons could identify the remaining GFRAL neurons and help elucidate whether GFRAL represents one or multiple populations of neurons.

2) The authors show that treatment with GDF15 induces Fos expression in CGRP neurons in the parabrachial nucleus (PBN) and this induction depends on CCK-expressing neurons (ablation study) and CCK (antagonist study). The implication of these studies is that CCK/GFRAL neurons project directly to CGRP-expressing neurons in the PBN and that CCK plays a significant role in the anorexia phenotype. It is a little surprising that CCK has such a large effect considering that the GFRAL/CCK neurons are probably also glutamatergic and may express other neuropeptides. Authors should determine whether CGRP neurons in the PBN express CCK receptors and whether they are type 1 and/or type 2 receptors. Demonstrating that CCK signaling in PBN is sufficient for the anorexia phenotype is also important.

Reviewer #2:

The manuscript by Beebe and colleagues investigates various aspects of GDF15 signaling and its receptor GFRAL within the brainstem that extends many previous findings about this cytokine system. This current study reports five main findings: (1) GFRAL-positive neurons in the AP and NTS partially overlap with CCK. (2) GDF produces negative affect, as measured by conditioned taste aversion, conditioned place aversion, and increased pica behavior. (3) GDF15 activates CCK neurons (as measured by Fos expression), as well as several downstream brain areas including the PBN and PVN. (4) Ablating CCK neurons in the brainstem or injecting mice with a global CCK antagonist attenuates the anorexigenic effects of GDF15. (5) GFRAL receptor blockade with a monoclonal antibody attenuates the adverse side effects of a cancer therapeutic drug, cisplatin.

Taken together, these results extend the findings of previous studies that reported the anorexigenic effects of GDF15 and the expression patterns of GFRAL in the brainstem. Many of the experiments in the study seem well done, however, some concerns dampen enthusiasm for this study.

1) The quantitative results of the Fos studies reported in Figure 1 and Supplementary Figure 1 are not sufficient to justify the conclusion that "GFRAL-positive neurons in the AP and NTS co-localize with CCK." Indeed, the only quantification provided indicates that less than half (45%) of GFRAL cells co-localize with CCK. Could the authors provide a Table with exact measurements across many animals to fully report the quantitative details of these experiments? If 45% of GFRAL cells co-localize with CCK, then why does the Venn diagram in Supplementary Figure 1 imply that much more than 45% (looks like 2/3) co-localize? Is this Venn diagram based on data, or only a rough approximation? What is the percentage of CCK neurons that colocalize with GFRAL? The number of details missing in this analysis calls into question any major conclusion. Certainly, the claim that "GFRAL is localized to CCK-positive neurons in the AP/NTS," as highlighted in the text, is not justified by the data.

2) Why is the Pica behavior performed in rats whereas all other experiments are performed in mice? Switching model organisms in the middle of a study seems odd. Wouldn't this experiment be relatively easy to perform in mice?

3) The title of one of the Results sections… "The effects of GDF15 are mediated by CCK neurons" is not actually demonstrated in this section. The section that follows reports the effects of GDF15 administration on Fos staining. One cannot conclude from these studies that the effects of GDF15 are mediated by CCK neurons. Ironically, this could be the conclusion of the following section, which demonstrates the necessity of CCK neural activity for GFF15-mediated anorexia.

4) Why do the authors study a loss of function of CCK neurons using two methods in Figure 4 (the Tobacco Etch Protease and CCK receptor antagonist), and then, when studying the effects of cisplatin, use a completely different method in Figure 5 (monoclonal antibody against GFRAL) to study GFRAL neurons instead of CCK neurons? This study is sometimes about CCK-expressing neurons and sometimes about GFRAL-expressing neurons, and some consistency across experiments would lend to better overall conclusions.

5) Does the ablation of CCK-expressing cells in the AP/NTS cause long-term changes in food intake behavior? The results in Figure 4 imply that ablation of these cells does not alter food intake, however, they prevent the anorexia caused by GDF15 administration. Could the authors make a larger conclusion about the fact that there is no result of CCK neuron ablation in the absence of GDF15 administration?

6) The histology results shown in Figure 5D are too small to discern.

Reviewer #3:

The paper provides novel insight into the population of brainstem neurons that express the receptor GFRAL for the cytokine GDF15 that is produced in pathological states and leads to anorexia and weight loss. In addition, they show data that suggest the effect of GDF15 to reduce food intake is in part mediated by CCK expressing neurons in the NTS and AP and by CCK signaling. Moreover, they show that GDF15 evoke conditioned taste and food aversion. These are important observations in a field of research that is rapidly expanding. Some concerns remain that would need to be addressed.

1) The animal models that have Cre-recombinase can have leaky Cre-expression. At the minimum, the authors should comment on the specificity of the Cre lines used for labeling the neuropeptidergic neurons studied.

2) The number of animals used in each experiment seems to be missing from all figures/experiments.

3) Given the importance of the anatomical characterizations of GFRAL neurons in the brainstem, it would be useful to include quantification of cells in Figure 1.

4) "Together these data suggest that GDF15 is not a natural satiety factor…". At the moment this statement is not very convincing, and the experiments were not designed to generate this conclusion.

5) Conditioned place aversion (Figure 2D). I might have missed something here, which could be clarified in the text, however it seems that in both pre-test and test conditions animals still prefer the side of the chamber in which they received GDF15, instead of switching the preference to the saline side. Can the decrease in preference be considered as a place aversion, even though the animals still prefer the same side of the chamber?

6) Figure 5G: the reduction in food intake by cisplatin in the presence of control Ab seems to be small and only significant at day 2 (?) Clarify statistics and N. In Supplementary Figure 5A and B would be helpful to see the correlation between GDF15 and cumulative food intake and %BW in control mice injected saline.

---

## [Author Response]

The major concerns that need to be addressed are as follows:1) The molecular identity of only about half of the GFRAL neurons was determined. What is the molecular identity of the other half? What fast transmitters do GFRAL neurons use, if any? Single-cell RNA-Seq is one way to answer these questions but is probably not feasible in 2 months unless the authors have already initiated this approach.

We were able to carry out two experiments before complete lockdown. We carried out some RNAScope in situ hybridisation, to further analyse the relationship between GFRAL and CCK. This showed that 69 ± 2 % of GFRAL neurons in the area postrema contain *Cck* mRNA, though the proportion of double-labelled cells is lower in the NTS (35 ± 4 %). As 46 ± 2 % of CCK neurons in the AP contain GFRAL, this represents a significant population. In addition, using a *Slc17a6*^Cre^::eYFP mouse we were able to show that GFRAL immunoreactive cells co-express *VGlut*. This fits with the consensus that CCK neurons in the brainstem are glutamatergic. Thus, we feel that this is ample evidence to confirm that GFRAL cells are predominantly CCK-ergic and have reworded the manuscript accordingly.

2) Addressing the identity of GFRAL post-synaptic neurons that mediate anorexia or other phenotypes is one of the goals of this study. GDF15 activates Fos in parabrachial CGRP neurons, in agreement with previous results, suggesting that GFRAL (CCK subtype) neurons may directly innervate CGRP neurons; however, experiments should be included that reveal whether the signaling is direct or indirect. Is the projection to the PBN the only one that mediates anorexia in response to GFRAL neuron activation?

We do not have a *Gfral*-cre mouse available. However, Roman et al., 2016, have demonstrated, using channel rhodopsin-assisted mapping, that CCK^NTS^ neurons project directly to CGRP^PBN^ cells. We will cite this paper explicitly on this point. Bearing in mind the limitations of tracing techniques, we have shown that the majority of CCK/GFRAL^NTS^ and CCK/GFRAL^AP^ neurons, which are activated by GDF15, project to the PBN and not to other identified, presumably transynaptically activated areas. Furthermore, when we ablate these PBN-projecting CCK neurons, the anorectic response to GDF15 is lost. Importantly, 59% of CCK neurons activated by GDF15 are in the AP, whereas only 11% are in the NTS. Unfortunately, it will be difficult to separate accurately the NTS and AP GFRAL populations with AAV injections, since they span the nuclei. In the manuscript, we have also tried to highlight the population of unidentified, GDF15-activated neurons in the NTS. These too may be dependent on CCK input.

It may be possible to establish the role of CGRP^PBN^ neurons in the response to GDF15, using *Calca*-cre mice, that can be injected with a cre-dependent AAV-caspase. However, due to Covid lockdown, we cannot initiate this experiment. We have tried to be clear that, while some of the neurons activated in the PBN contain *Calca*-cre, not all of them do. Therefore, we are not convinced that we would necessarily block the whole effect of GDF15 by inactivating CGRP neurons (see point below also). We have re-written the paragraph to make it clear that some of the activated neurons in the PBN contain CGRP, and so these are one downstream target.

3) The observation that CCK neurons and CCK itself are an important contributor to the anorexic phenotype is an interesting observation. If the GFRAL/CCK neurons are glutamatergic, it is surprising that CCK is so important. Experiments showing the CCK action in the PBN is important and determining which CCK receptors are involved are needed to solidify this interesting observation.

This relates to the point above. We are reliant on the *Calca*-cre mouse injected with AAV-mCherry for identification of CGRP^PBN^ neurons. Although we, and some others, have attempted immunostaining for native CGRP and CCK1R, the results are not reliable and we would not trust interpretation based on this methodology. Thus, we also have performed dual label in situ hybridisation histology for *Calca* and *Cckr1*, which clearly shows a small proportion of CGRP neurons contain *Cckr1* mRNA.

Roman et al., 2016, showed that the immediate electrical response to stimulation of CCK^NTS^ neurons can be blocked with glutamate antagonists. Although establishing that input, it does not rule out an effect of CCK on the functioning of CGRP^PBN^ or other PBN cells. Our results demonstrate that about half of the acute effects of GDF15 are dependent on CCK receptor signalling. We have rewritten the section to incorporate our new findings and to put the size of the effect of the antagonist into context. We agree that the effect of the CCK antagonist is significant, and we cannot discount an important role for synaptically activated cells in the NTS (a region which also contains CCKR1). We are keen to investigate this signalling pathway further, but cannot because of Covid restrictions.

4) Does CCK ablation in AP/NTS have long-term effects on food intake or body weight. Perhaps this experiment has already been published. If so, provide a reference.

To our knowledge, the long-term effects of ablating CCK neurons has not been published. Across the time frame we have studied in our mice, we have not seen any obvious changes in body weight. This has now been included in Figure 4—figure supplement 1.

5) The conditioned place aversion experiment needs clarification. The authors say that testing was the same as day one, i.e. 30-min (1800-s) session. But Figure 2D shows only ~500 s on the preferred side which is less than half of the total time. Pairing GDF15 with the preferred side decreased that time to about 250 seconds suggesting that it is aversive. One problem is that "CPA score" is never defined.

Thank you for this observation. We have not made this point clear, but there is an adjoining “corridor” between the two chambers, permitting “unforced choice.” This is considered good practice in any form of place conditioning, as described by Prus et al., (2009; PMID: 21204336). As the mice are free to explore three areas, the figures do not appear to add up. Thus, we have re-graphed the results to show the absolute time spent in the GDF15-paired chamber, which we hope is more clear. We will clarify this in the text and removed reference to the CPA score. Please see a fuller explanation in response to the reviewer.

6) The number of mice used for each experiment should be included in the figure legends.

Numbers of mice will be included in each figure legend.

The reviewers also have specific suggestions for improvement of this paper which can be found in the individual comments.Reviewer #1:The authors of this paper provide data demonstrating that GDF15 induces anorexia and nausea-like behavior in mice/rats and suggest that GDF15 is not a physiological regulator of satiety but rather a sickness signal. They also show that cisplatin stimulates GDF15 expression and the anorexia effect of this cancer-treatment drug depends on GFRAL. Many of the experiments are similar to those recently published by some of the same authors in Cell Metabolism (2020, 31:1-12).

This is not our paper, and it does not show the same data. However, it has been cited.

1) The strength of this paper is that the authors begin to characterize the signaling molecules expressed by GFRAL neurons and the neural circuits that they activate. Using a candidate-gene approach they provide evidence that about half of the GFRAL neurons express CCK and/or TH, but the remaining neurons remain unidentified. What fraction of the GFRAL neurons are glutamatergic? Using RNA-Seq of GFRAL-expressing neurons could identify the remaining GFRAL neurons and help elucidate whether GFRAL represents one or multiple populations of neurons.

As described above, we have now provided additional data to show GFRAL neurons contain *VGlut*. Importantly, RNAScope shows that the majority of *Gfral* cells contain *Cck*, and so they are the predominant phenotype. We do not have the opportunity, resources or time to carry out RNASeq analysis.

2) The authors show that treatment with GDF15 induces Fos expression in CGRP neurons in the parabrachial nucleus (PBN) and this induction depends on CCK-expressing neurons (ablation study) and CCK (antagonist study). The implication of these studies is that CCK/GFRAL neurons project directly to CGRP-expressing neurons in the PBN and that CCK plays a significant role in the anorexia phenotype. It is a little surprising that CCK has such a large effect considering that the GFRAL/CCK neurons are probably also glutamatergic and may express other neuropeptides. Authors should determine whether CGRP neurons in the PBN express CCK receptors and whether they are type 1 and/or type 2 receptors. Demonstrating that CCK signaling in PBN is sufficient for the anorexia phenotype is also important.

We have now shown that a proportion of CGRP^PBN^ neurons contain *Cckr1* mRNA and have stated clearly that there may be other GFRAL targets in the PBN. We have pointed out that about half of the acute anorectic effect of GDF15 is blocked by the CCK receptor antagonist. Importantly, we reiterate that GFRAL neurons appear to activate unidentified neurons in the both the PBN and in the NTS.

Reviewer #2:[…] 1) The quantitative results of the Fos studies reported in Figure 1 and Supplementary Figure 1 are not sufficient to justify the conclusion that "GFRAL-positive neurons in the AP and NTS co-localize with CCK." Indeed, the only quantification provided indicates that less than half (45%) of GFRAL cells co-localize with CCK. Could the authors provide a Table with exact measurements across many animals to fully report the quantitative details of these experiments?

We now have provided the exact measurements in a supplementary table (Supplementary file 1). This includes new data from the RNAScope study, which indicates a higher co-localisation.

If 45% of GFRAL cells co-localize with CCK, then why does the Venn diagram in Supplementary Figure 1 imply that much more than 45% (looks like 2/3) co-localize? Is this Venn diagram based on data, or only a rough approximation? What is the percentage of CCK neurons that colocalize with GFRAL? The number of details missing in this analysis calls into question any major conclusion. Certainly, the claim that "GFRAL is localized to CCK-positive neurons in the AP/NTS," as highlighted in the text, is not justified by the data.

We have now dealt with this by carrying out dual-label RNAScope (see comments to editors). This clearly shows that a majority of *Gfral*-containing neurons also contain *Cck* mRNA. Likewise, 46 % of *Cck* neurons in the AP contain *Gfral*. These data, along with additional quantification of co-localisation of *VGlut* and TH in GFRAL neurons has also been included in the supplementary table (Supplementary file 1). Although we had stated that the Venn diagram was only a rough representation, we agree that it does not portray our results accurately, so we have now omitted it from the manuscript.

2) Why is the Pica behavior performed in rats whereas all other experiments are performed in mice? Switching model organisms in the middle of a study seems odd. Wouldn't this experiment be relatively easy to perform in mice?

Pica behaviour is notoriously difficult to perform in mice as they have a tendency to gnaw at kaolin and spread the powder liberally around their cages, making accurate weighing almost impossible. We think that adding data from another species adds value, but we decided only to include the figure as supplementary material.

3) The title of one of the Results sections "The effects of GDF15 are mediated by CCK neurons" is not actually demonstrated in this section. The section that follows reports the effects of GDF15 administration on Fos staining. One cannot conclude from these studies that the effects of GDF15 are mediated by CCK neurons. Ironically, this could be the conclusion of the following section, which demonstrates the necessity of CCK neural activity for GFF15-mediated anorexia.

Strictly true, but as we have now shown GFRAL is expressed mostly in CCK neurons, it is a fair assumption. That said, we have altered the sub-section title to “GDF15 activates CCK^AP/NTS^ neurons.”

4) Why do the authors study a loss of function of CCK neurons using two methods in Figure 4 (the Tobacco Etch Protease and CCK receptor antagonist), and then, when studying the effects of cisplatin, use a completely different method in Figure 5 (monoclonal antibody against GFRAL) to study GFRAL neurons instead of CCK neurons? This study is sometimes about CCK-expressing neurons and sometimes about GFRAL-expressing neurons, and some consistency across experiments would lend to better overall conclusions.

With respect, the reviewer will not have seen our additional results, which shows the high overlap between GFRAL and CCK neurons. Thus, we are then trying to study the relevance of that cell mediator in different ways. We believe that providing evidence using different methods can add strength to the argument. Caspase will kill *Cck*^Cre^ neurons, importantly including GFRAL-expressing CCK neurons, as we have verified in *post mortem* cell counts (Figure 4—figure supplement 1). CCK neurons in the brainstem are glutamatergic and may contain other transmitters. Therefore, we wanted to assess the role of CCK itself (as opposed to other transmitters) in mediating the effects of GDF15, and so used a receptor antagonist. The last section of the paper addresses a separate question of whether the GFRAL receptor can be targeted in translational research. The GFRAL receptor antibody is an excellent tool to provide useful data, as an antagonist is not available.

5) Does the ablation of CCK-expressing cells in the AP/NTS cause long-term changes in food intake behavior? The results in Figure 4 imply that ablation of these cells does not alter food intake, however, they prevent the anorexia caused by GDF15 administration. Could the authors make a larger conclusion about the fact that there is no result of CCK neuron ablation in the absence of GDF15 administration?

To our knowledge, the long-term effects of ablating CCK neurons has not been published. Across the time frame we have studied in our mice, we have not seen any obvious changes in body weight. This has now been included as Figure 4—figure supplement 1. We are not in a position to speculate whether CCK neurons have a homeostatic role, as our experiments have not been designed specifically to answer such a question.

6) The histology results shown in Figure 5D are too small to discern.

We have remade this figure.

Reviewer #3:[…] 1) The animal models that have Cre-recombinase can have leaky Cre-expression. At the minimum, the authors should comment on the specificity of the Cre lines used for labeling the neuropeptidergic neurons studied.

References are provided in the Materials and methods section for all the mouse lines used, which demonstrate that each produces a true representation of endogenous gene expression. It is known that both the *Cck*^Cre^ and *Calca*^Cre^ lines may report on more cells than express the gene in adult mice; suggesting that more cells may express significant levels of these genes during development. This is a fair point regarding the initial section, where we used a candidate-gene approach and transgenic mouse lines. However, we have now followed this up with dual-label RNAScope in adult mice, which shows a good consensus. For the experiment using *Calca*^Cre^, there is not an issue, since adult mice were injected with a viral reporter.

2) The number of animals used in each experiment seems to be missing from all figures/experiments.

Numbers have been included

3) Given the importance of the anatomical characterizations of GFRAL neurons in the brainstem, it would be useful to include quantification of cells in Figure 1.

Quantification has been added in the new supplementary table (Supplementary file 1).

4) "Together these data suggest that GDF15 is not a natural satiety factor…". At the moment this statement is not very convincing, and the experiments were not designed to generate this conclusion.

We have added reference to the recent publications and tempered the sentence by saying “Together, these and recently published data (Borner et al., 2020 two papers) suggest that GDF15 is probably not a satiety factor.” The following sentence in our manuscript also provides another four references which contain additional supporting evidence for this statement. In this manuscript we go on to show that GFRAL neurons are only activated by stimuli which cause gastric malaise, and not by stimuli that induce normal satiety. We also show that treatment of mice with the GFRAL monoclonal antibody itself does not affect feeding or body weight, but does block anorexia caused by a drug that causes gastric malaise.

5) Conditioned place aversion (Figure 2D). I might have missed something here, which could be clarified in the text, however it seems that in both pre-test and test conditions animals still prefer the side of the chamber in which they received GDF15, instead of switching the preference to the saline side. Can the decrease in preference be considered as a place aversion, even though the animals still prefer the same side of the chamber?

Thank you for this observation, as we can see how confusing this is. It highlights the issue of using a CPA “score” to denote aversion. As done by others, CPA was calculated by [time in chamber A] minus [time in chamber B] and, therefore, represents the *difference* in time spent between chambers, not the absolute time spent in each chamber. This calculation is often used in CPP/CPA experiments to account for the inherent preference of mice to one-side of a two-chamber apparatus and normalises this preference when there is a difference in separate treatment groups (which is not an issue here). Our study is further complicated by the fact that we have an adjoining corridor between two chambers, permitting unforced choice (as described by Prus and colleagues, PMID 21204336), but giving the mice three areas to explore.

We accept that the CPA score is misleading and perhaps not appropriate here. Therefore, we have changed Figure 2C to depict time spent in the GDF15-paired chamber and have described explicitly both our apparatus and analysis in the Materials and methods section.

6) Figure 5G: the reduction in food intake by cisplatin in the presence of control Ab seems to be small and only significant at day 2 (?) Clarify statistics and N. In Supplementary Figure 5A and B would be helpful to see the correlation between GDF15 and cumulative food intake and %BW in control mice injected saline.

Thank you for this comment. We agree that the result for day 1 looks small; this is because the first injection of cisplatin is given on day 2. We have altered the axis in the Figure 5G to reflect this (i.e. daily not cumulative food intake). Stats and N are now clear in the legend. We have included the correlation analysis for mice injected with control in Figure 5—figure supplement 1.